# A consensus guide to capturing the ability to inhibit actions and impulsive behaviors in the stop-signal task

Frederick Verbruggen[1]*, Adam R Aron[2], Guido PH Band[3], Christian Beste[4], Patrick G Bissett[5], Adam T Brockett[6], Joshua W Brown[7], Samuel R Chamberlain[8], Christopher D Chambers[9], Hans Colonius[10], Lorenza S Colzato[3], Brian D Corneil[11], James P Coxon[12], Annie Dupuis[13], Dawn M Eagle[8], Hugh Garavan[14], Ian Greenhouse[15], Andrew Heathcote[16], René J Huster[17], Sara Jahfari[18], J Leon Kenemans[19], Inge Leunissen[20], Chiang-Shan R Li[21], Gordon D Logan[22], Dora Matzke[23], Sharon Morein-Zamir[24], Aditya Murthy[25], Martin Paré[26], Russell A Poldrack[5], K Richard Ridderinkhof[23], Trevor W Robbins[8], Matthew Roesch[6], Katya Rubia[27], Russell J Schachar[13], Jeffrey D Schall[22], Ann-Kathrin Stock[4], Nicole C Swann[15], Katharine N Thakkar[28], Maurits W van der Molen[23], Luc Vermeylen[1], Matthijs Vink[19], Jan R Wessel[29], Robert Whelan[30], Bram B Zandbelt[31], C Nico Boehler[1]

[1]Experimental Psychology, Ghent University, Ghent, Belgium; [2]University of California, San Diego, San Diego, United States; [3]Leiden University, Leiden, Netherlands; [4]Dresden University of Technology, Dresden, Germany; [5]Stanford University, Stanford, United States; [6]University of Maryland, College Park, United States; [7]Indiana University, Bloomington, United States; [8]University of Cambridge, Cambridge, United Kingdom; [9]Cardiff University, Cardiff, United Kingdom; [10]Oldenburg University, Oldenburg, Germany; [11]University of Western Ontario, London, Canada; [12]Monash University, Clayton, Australia; [13]University of Toronto, Toronto, Canada; [14]University of Vermont, Burlington, United States; [15]University of Oregon, Eugene, United States; [16]University of Tasmania, Hobart, Australia; [17]University of Oslo, Oslo, Norway; [18]Spinoza Centre Amsterdam, Amsterdam, Netherlands; [19]Utrecht University, Utrecht, Netherlands; [20]KU Leuven, Leuven, Belgium; [21]Yale University, New Haven, United States; [22]Vanderbilt University, Nashville, United States; [23]University of Amsterdam, Amsterdam, Netherlands; [24]Anglia Ruskin University, Cambridge, United Kingdom; [25]Indian Institute of Science, Bangalore, India; [26]Queen's University, Kingston, Canada; [27]King's College London, London, United Kingdom; [28]Michigan State University, East Lansing, United States; [29]University of Iowa, Iowa City, United States; [30]Trinity College Dublin, Dublin, Ireland; [31]Donders Institute, Nijmegen, Netherlands

*For correspondence:
frederick.verbruggen@ugent.be

**Abstract** Response inhibition is essential for navigating everyday life. Its derailment is considered integral to numerous neurological and psychiatric disorders, and more generally, to a wide range of behavioral and health problems. Response-inhibition efficiency furthermore correlates with treatment outcome in some of these conditions. The stop-signal task is an essential tool to determine how quickly response inhibition is implemented. Despite its apparent simplicity, there are many features (ranging from task design to data analysis) that vary across studies in ways that can easily compromise the validity of the obtained results. Our goal is to facilitate a more accurate use of the stop-signal task. To this end, we provide 12 easy-to-implement consensus

recommendations and point out the problems that can arise when they are not followed. Furthermore, we provide user-friendly open-source resources intended to inform statistical-power considerations, facilitate the correct implementation of the task, and assist in proper data analysis.
DOI: https://doi.org/10.7554/eLife.46323.001

## Introduction

The ability to suppress unwanted or inappropriate actions and impulses ('response inhibition') is a crucial component of flexible and goal-directed behavior. The stop-signal task (*Lappin and Eriksen, 1966*; *Logan and Cowan, 1984*; *Vince, 1948*) is an essential tool for studying response inhibition in neuroscience, psychiatry, and psychology (among several other disciplines; see Appendix 1), and is used across various human (e.g. clinical vs. non-clinical, different age groups) and non-human (primates, rodents, etc.) populations. In this task, participants typically perform a go task (e.g. press left when an arrow pointing to the left appears, and right when an arrow pointing to the right appears), but on a minority of the trials, a stop signal (e.g. a cross replacing the arrow) appears after a variable stop-signal delay (SSD), instructing participants to suppress the imminent go response (*Figure 1*). Unlike the latency of go responses, response-inhibition latency cannot be observed directly (as successful response inhibition results in the absence of an observable response). The stop-signal task is unique in allowing the estimation of this covert latency (stop-signal reaction time or SSRT; *Box 1*). Research using the task has revealed links between inhibitory-control capacities and a wide range of behavioral and impulse-control problems in everyday life, including attention-deficit/hyperactivity disorder, substance abuse, eating disorders, and obsessive-compulsive behaviors (for meta-analyses, see e.g. *Bartholdy et al., 2016*; *Lipszyc and Schachar, 2010*; *Smith et al., 2014*).

Today, the stop-signal field is flourishing like never before (see Appendix 1). There is a risk, however, that the task falls victim to its own success, if it is used without sufficient regard for a number of important factors that jointly determine its validity. Currently, there is considerable heterogeneity in how stop-signal studies are designed and executed, how the SSRT is estimated, and how results of stop-signal studies are reported. This is highly problematic. First, what might seem like small design details can have an immense impact on the nature of the stop process and the task. The heterogeneity in designs also complicates between-study comparisons, and some combinations of design and analysis features are incompatible. Second, SSRT estimates are unreliable when inappropriate estimation methods are used or when the underlying race-model assumptions are (seriously) violated (see *Box 1* for a discussion of the race model). This can lead to artefactual and plainly incorrect results. Third, the validity of SSRT can be checked only if researchers report all relevant methodological information and data.

Here, we aim to address these issues by consensus. After an extensive consultation round, the authors of the present paper agreed on 12 recommendations that should safeguard and further improve the overall quality of future stop-signal research. The recommendations are based on previous methodological studies or, where further empirical support was required, on novel simulations (which are reported in Appendices 2–3). A full overview of the stop-signal literature is beyond the scope of this study (but see e.g. *Aron, 2011*; *Bari and Robbins, 2013*; *Chambers et al., 2009*; *Schall et al., 2017*; *Verbruggen and Logan, 2017*, for comprehensive overviews of the clinical, neuroscience, and cognitive stop-signal domains; see also the meta-analytic reviews mentioned above).

Below, we provide a concise description of the recommendations. We briefly introduce all important concepts in the main manuscript and the boxes. Appendix 4 provides an additional systematic overview of these concepts and their common alternative terms. Moreover, this article is accompanied by novel open-source resources that can be used to execute a stop-signal task and analyze the resulting data, in an easy-to-use way that complies with our present recommendations (https://osf.io/rmqaw/). The source code of the simulations (Appendices 2–3) is also provided, and can be used in the planning stage (e.g. to determine the required sample size under varying conditions, or acceptable levels of go omissions and RT distribution skew).

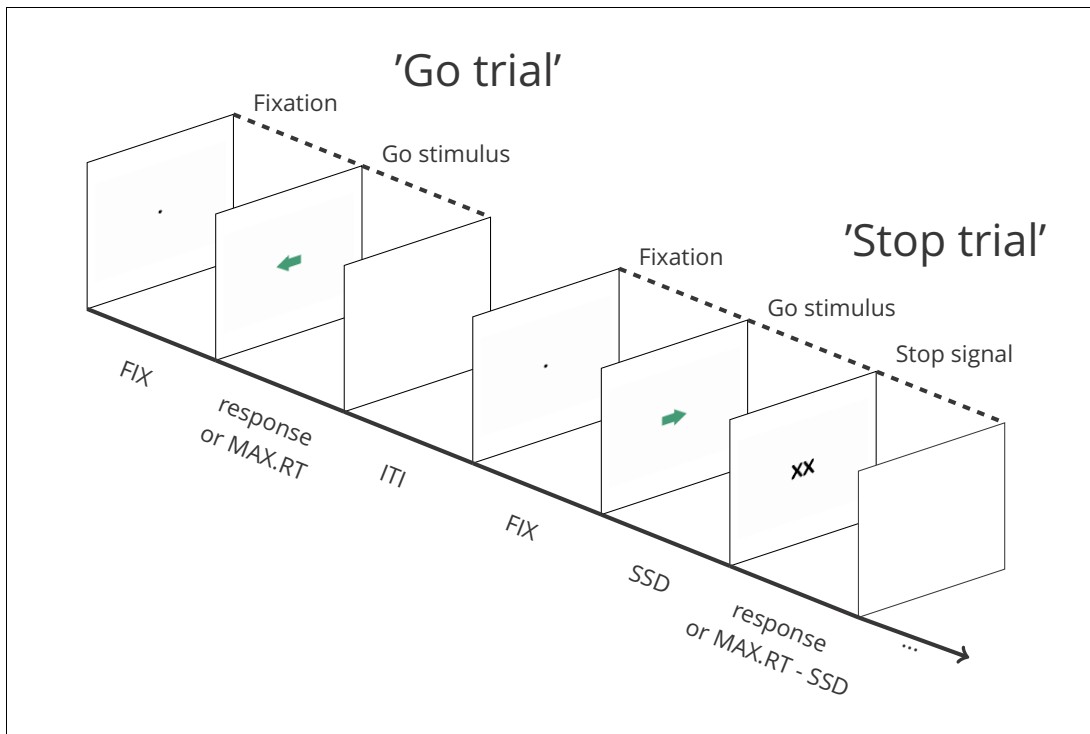

**Figure 1.** Depiction of the sequence of events in a stop-signal task (see https://osf.io/rmqaw/ for open-source software to execute the task). In this example, participants respond to the direction of green arrows (by pressing the corresponding arrow key) in the go task. On one fourth of the trials, the arrow is replaced by 'XX' after a variable stop-signal delay (FIX = fixation duration; SSD = stop signal delay; MAX.RT = maximum reaction time; ITI = intertrial interval).

DOI: https://doi.org/10.7554/eLife.46323.002

## Results and discussion

The following recommendations are for stop-signal users who are primarily interested in obtaining a reliable SSRT estimate under standard situations. The stop-signal task (or one of its variants) can also be used to study various aspects of executive control (e.g. performance monitoring, strategic adjustments, or learning) and their interactions, for which the design might have to be adjusted. However, researchers should be aware that this will come with specific challenges (e.g. *Bissett and Logan, 2014*; *Nelson et al., 2010*; *Verbruggen et al., 2013*; *Verbruggen and Logan, 2015*).

### How to design stop-signal experiments

#### Recommendation 1: Use an appropriate go task

Standard two-choice reaction time tasks (e.g. in which participants have to discriminate between left and right arrows) are recommended for most purposes and populations. When very simple go tasks are used, the go stimulus and the stop signal will closely overlap in time (because the SSD has to be very short to still allow for the possibility to inhibit a response), leading to violations of the race model as stop-signal presentation might interfere with encoding of the go stimulus. Substantially increasing the difficulty of the go task (e.g. by making the discrimination much harder) might also influence the stop process (e.g. the underlying latency distribution or the probability that the stop process is triggered). Thus, very simple and very difficult go tasks should be avoided unless the researcher has theoretical or methodological reasons for using them (for example, simple detection tasks have been used in animal studies. To avoid responses before the go stimulus is presented or close overlap between the presentation of go stimulus and stop signal, the intertrial interval can be drawn from a random exponential distribution. This will make the occurrence of the go stimulus unpredictable, discouraging anticipatory responses). While two-choice tasks are the most common, we note that the 'anticipatory response' variant of the stop-signal task (in which participants have to

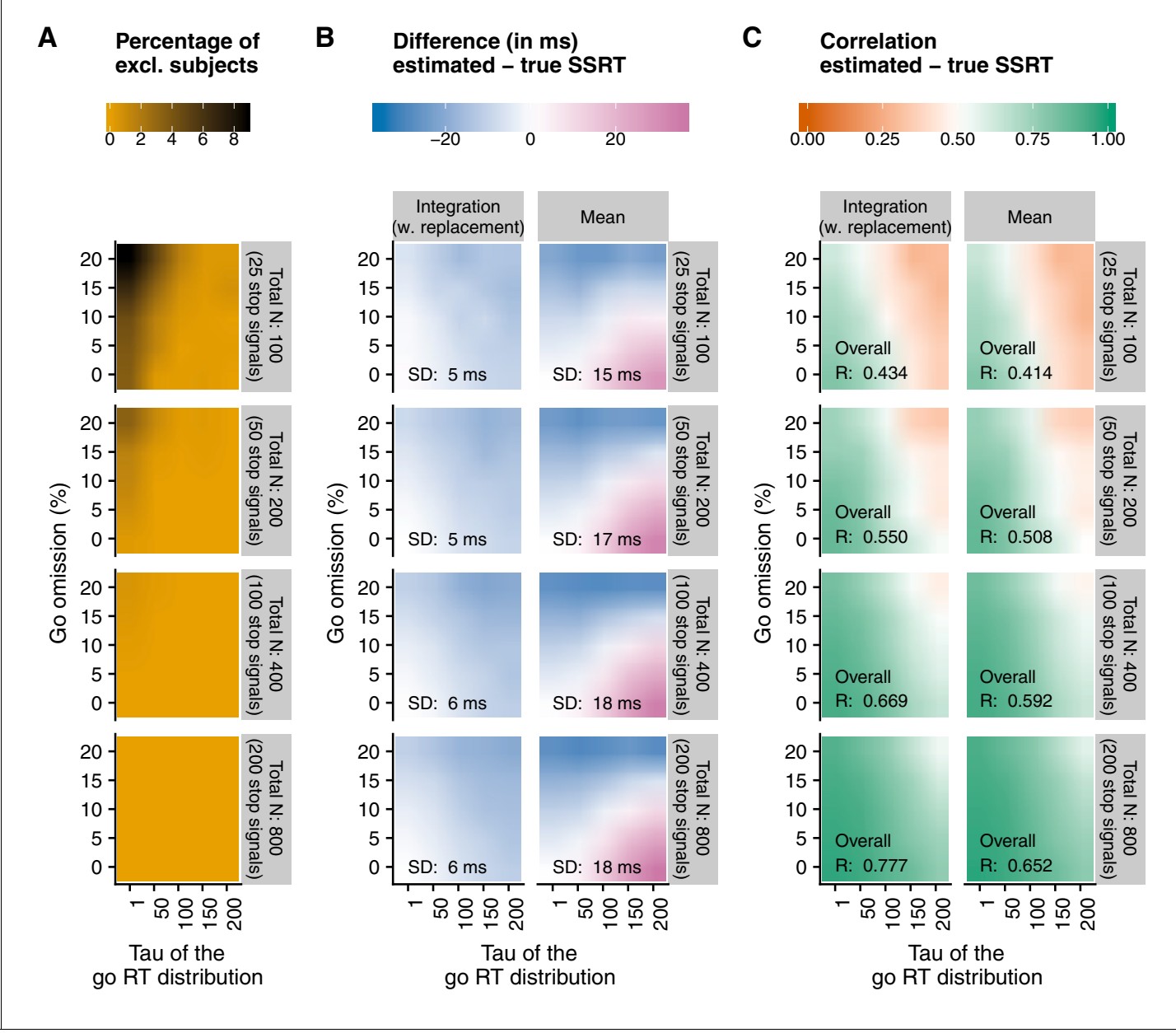

**Figure 2.** Main results of the simulations reported in Appendix 2. Here, we show a comparison of the integration method (with replacement of go omissions) and the mean method, as a function of percentage of go omissions, skew of the RT distribution ($\tau_{go}$), and number of trials. Appendix 2 provides a full overview of all methods. (A) The number of excluded 'participants' (RT on unsuccessful stop trials > RT on go trials). As this check was performed before SSRTs were estimated (see Recommendation 7), the number was the same for both estimation methods. (B) The average difference between the estimated and true SSRT (positive values = overestimation; negative values = underestimation). SD = standard deviation of the difference scores (per panel). (C) Correlation between the estimated and true SSRT (higher values = more reliable estimate). Overall R = correlation when collapsed across percentage of go omissions and $\tau_{go}$. Please note that the overall correlation does not necessarily correspond to the average of individual correlations.

DOI: https://doi.org/10.7554/eLife.46323.008

press a key when a moving indicator reaches a stationary target) also holds promise (e.g. *Leunissen et al., 2017*).

## Box 1. The independent race model

Here, we provide a brief discussion of the independent race model, without the specifics of the underlying mathematical basis. However, we recommend that stop-signal users read the original modelling papers (e.g. *Logan and Cowan, 1984*) to fully understand the task and the main behavioral measures, and to learn more about variants of the race model (e.g. *Boucher et al., 2007*; *Colonius and Diederich, 2018*; *Logan et al., 2014*; *Logan et al., 2015*).

Response inhibition in the stop-signal task can be conceptualized as an independent race between a 'go runner', triggered by the presentation of a go stimulus, and a 'stop runner', triggered by the presentation of a stop signal (*Logan and Cowan, 1984*). When the 'stop runner' finishes before the 'go runner', response inhibition is successful and no response is emitted (*successful stop trial*); but when the 'go runner' finishes before the 'stop runner', response inhibition is unsuccessful and the response is emitted (*unsuccessful stop trial*). The independent race model mathematically relates (a) the latencies (RT) of responses on unsuccessful stop trials; (b) RTs on go trials; and (c) the probability of responding on stop trials [p(respond| stop signal)] as a function of stop-signal delay (yielding 'inhibition functions'). Importantly, the independent race model provides methods for estimating the covert latency of the stop process (stop-signal reaction time; SSRT). These estimation methods are described in Materials and methods.

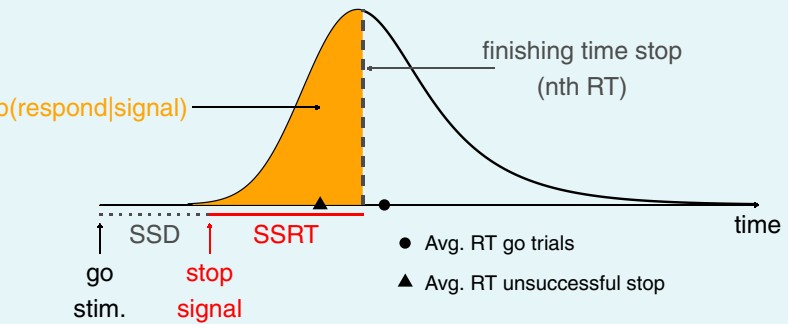

**Box 1—figure 1.** The independent race between go and stop.

DOI: https://doi.org/10.7554/eLife.46323.004

DOI: https://doi.org/10.7554/eLife.46323.003

## Recommendation 2: Use a salient stop signal

SSRT is the overall latency of a chain of processes involved in stopping a response, including the detection of the stop signal. Unless researchers are specifically interested in such perceptual or attentional processes, salient, easily detectable stop signals should be used (when auditory stop signals are used, these should not be too loud either, as very loud (i.e. >80 dB) auditory stimuli may produce a startle reflex). Salient stop signals will reduce the relative contribution of perceptual (afferent) processes to the SSRT, and the probability that within- or between-group differences can be attributed to them. Salient stop signals might also reduce the probability of a 'trigger failures' on stop trials (see *Box 2*).

## Recommendation 3: Present stop signals on a minority of trials

When participants strategically wait for a stop signal to occur, the nature of the stop-signal process and task change (complicating the comparison between conditions or groups; e.g. SSRT group differences might be caused by differential slowing or strategic adjustments). Importantly, SSRT estimates will also become less reliable when participants wait for the stop signal to occur

(*Verbruggen et al., 2013*, see also *Figure 2* and Appendix 2). Such waiting strategies can be discouraged by reducing the overall probability of a stop signal. For standard stop-signal studies, 25% stop signals is recommended. When researchers prefer a higher percentage of stop signals, additional measures to minimize slowing are required (see Recommendation 5).

### Recommendation 4: Use the tracking procedure to obtain a broad range of stop-signal delays

If participants can predict when a stop signal will occur within a trial, they might also wait for it. Therefore, a broad range of SSDs is required. The stop-signal delay can be continuously adjusted via a standard adaptive tracking procedure: SSD increases after each successful stop, and decreases after each unsuccessful stop; this converges on a probability of responding [p(respond|signal)] ≈ 0.50. Many studies adjust SSD in steps of 50 ms (which corresponds to three screen 'refreshes' for 60 Hz monitors). When step size is too small (for example 16 ms) the tracking may not converge in short experiments, whereas it may not be sensitive enough if step size is too large. Importantly, SSD should decrease after *all* responses on unsuccessful stop trials; this includes premature responses on unsuccessful stop trials (i.e. responses executed before the stop signal was presented) and choice errors on unsuccessful stop trials (e.g. when a left go response would have been executed on the stop trial depicted in *Figure 1*, even though the arrow was pointing to the right).

An adaptive tracking procedure typically results in a sufficiently varied set of SSD values. An additional advantage of the tracking procedure is that fewer stop trials are required to obtain a reliable SSRT estimate (*Band et al., 2003*). Thus, the tracking procedure is recommended for standard applications.

### Recommendation 5: Instruct participants not to wait and include block-based feedback

In human studies, task instructions should also be used to discourage waiting. At the very least, participants should be told that '*[they] should respond as quickly as possible to the go stimulus and not wait for the stop signal to occur*' (or something along these lines). To adults, the tracking procedure (if used) can also be explained to further discourage a waiting strategy (i.e. inform participants that the probability of an unsuccessful stop trial will approximate 0.50, and that SSD will increase if they gradually slow their responses).

Inclusion of a practice block in which adherence to instructions is carefully monitored is recommended. In certain populations, such as young children, it might furthermore be advisable to start with a practice block without stop signals to emphasize the importance of the go component of the task.

Between blocks, participants should also be reminded about the instructions. Ideally, this is combined with block-based feedback, informing participants about their mean RT on go trials, number of go omissions (with a reminder that this should be 0), and p(respond|signal) (with a reminder that this should be close to .50). The feedback could even include an explicit measure of response slowing.

### Recommendation 6: Include sufficient trials

The number of stop trials varies widely between studies. Our novel simulation results (see *Figure 2* and Appendix 2) indicate that reliable and unbiased SSRT group-level estimates can be obtained with 50 stop trials (with 25% stop signals in an experiment, this amounts to 200 trials in total. Usually, this corresponds to an experiment of 7–10 min including breaks), but only under 'optimal' or very specific circumstances (e.g. when the probability of go omissions is low and the go-RT distribution is not strongly skewed). Lower trial numbers (here we tested 25 stop trials) rarely produced reliable SSRT estimates (and the number of excluded subjects was much higher; see *Figure 2*). Thus, as a general rule of thumb, we recommend to have at least 50 stop trials for standard group-level comparisons. However, it should again be stressed that this may not suffice to obtain reliable individual estimates (which are required for e.g. individual-differences research or diagnostic purposes).

Thus, our simulations reported in Appendix 2 suggest that reliability increases with number of trials. However, in some clinical populations, adding trials may not always be possible (e.g. when patients cannot concentrate for a sufficiently long period of time), and might even be

## Box 2. Failures to trigger the stop process

The race model assumes that the go runner is triggered by the presentation of the go stimulus, and the stop runner by the presentation of the stop signal. However, go omissions (i.e. go trials without a response) are often observed in stop-signal studies. Our preferred SSRT method compensates for such go omissions (see Materials and methods). However, turning to the stopping process, studies using fixed SSDs have found that p(respond|signal) at very short delays (including SSD = 0 ms, when go and stop are presented together) is not always zero; this finding indicates that the stop runner may also not be triggered on all stop trials ('trigger failures').

The non-parametric estimation methods described in Materials and methods (see also Appendix 2) will overestimate SSRT when trigger failures are present on stop trials (*Band et al., 2003*). Unfortunately, these estimation methods cannot determine the presence or absence of trigger failures on stop trials. In order to diagnose in how far trigger failures are present in their data, researchers can include extra stop signals that occur at the same time of the go stimulus (i.e. SSD = 0, or shortly thereafter). Note that this number of zero-SSD trials should be sufficiently high to detect (subtle) within- or between-group differences in trigger failures. Furthermore, p(respond|signal) should be reported separately for these short-SSD trials, and these trials should not be included when calculating mean SSD or estimating SSRT (see Recommendation one for a discussion of problems that arise when SSDs are very short. Note that the (neural) mechanisms involved in stopping might also partly differ when SSD = 0; see for example *Swick et al., 2011*). Alternatively, researchers can use a parametric method to estimate SSRT. Such methods describe the whole SSRT distribution (unlike the non-parametric methods that estimate summary measures, such as the mean stop latency). Recent variants of such parametric methods also provide an estimate of the probability of trigger failures on stop trials (for the most recent version and specialized software, see *Matzke et al., 2019*).

DOI: https://doi.org/10.7554/eLife.46323.005

counterproductive (as strong fluctuations over time can induce extra noise). Our simulations reported in Appendix 3 show that for standard group-level comparisons, researchers can compensate for lower trial numbers by increasing sample size. Above all, we strongly encourage researchers to make informed decisions about number of trials and participants, aiming for sufficiently powered studies. The accompanying open-source simulation code can be used for this purpose.

## When and how to estimate SSRT

Recommendation 7: Do not estimate the SSRT when the assumptions of the race model are violated

SSRTs can be estimated based on the independent race model, which assumes an independent race between a go and a stop runner (*Box 1*). When this independence assumption is (seriously) violated, SSRT estimates become unreliable (*Band et al., 2003*). Therefore, the assumption should be checked. This can be done by comparing the mean RT on unsuccessful stop trials with the mean RT on go trials. Note that this comparison should include all trials with a response (including choice errors and premature responses), and it should be done for each participant and condition separately. SSRT should not be estimated when RT on unsuccessful stop trials is numerically longer than RT on go trials (see also, *Appendix 2—table 1*). More formal and in-depth tests of the race model can be performed (e.g. examining probability of responding and RT on unsuccessful stop trials as a function of delay); however, a large number of stop trials is required for such tests to be meaningful and reliable.

## Box 3. Check-lists for reporting stop-signal studies

The description of every stop-signal study should include the following information:

- Stimuli and materials
    - Properties of the go stimuli, responses, and their mapping
    - Properties of the stop signal
    - Equipment used for testing
- The procedure
    - The number of blocks (including practice blocks)
    - The number of go and stop trials per block
    - Detailed description of the randomization (e.g. is the order of go and stop trials fully randomized or pseudo-randomized?)
    - Detailed description of the tracking procedure (including start value, step size, minimum and maximum value) or the range and proportion of fixed stop-signal delays.
    - Timing of all events. This can include intertrial intervals, fixation intervals (if applicable), stimulus-presentation times, maximum response latency (and whether a trial is terminated when a response is executed or not), feedback duration (in case immediate feedback is presented), etc.
    - A summary of the instructions given to the participant, and any feedback-related information (full instructions can be reported in Supplementary Materials).
    - Information about training procedures (e.g. in case of animal studies)
- The analyses
    - Which trials were included when analyzing go and stop performance
    - Which SSRT estimation method was used (see Materials and methods), providing additional details on the exact approach (e.g. whether or not go omissions were replaced; how go and stop trials with a choice errors–e.g. left response for right arrows–were handled; how the nth quantile was estimated; etc.)
    - Which statistical tests were used for inferential statistics

Stop-signal studies should also report the following descriptive statistics for each group and condition separately (see Appendix 4 for a description of all labels):

- Probability of go omissions (no response)
- Probability of choice errors on go trials
- RT on go trials (mean or median). We recommend to report intra-subject variability as well (especially for clinical studies).
- Probability of responding on a stop trial (for each SSD when fixed delays are used)
- Average stop-signal delay (when the tracking procedure is used); depending on the set-up, it is advisable to report (and use) the 'real' SSDs (e.g. for visual stimuli, the requested SSD may not always correspond to the real SSD due to screen constraints).
- Stop-signal reaction time
- RT of go responses on unsuccessful stop trials

DOI: https://doi.org/10.7554/eLife.46323.006

## Recommendation 8: If using a non-parametric approach, estimate SSRT using the integration method (with replacement of go omissions)

Different SSRT estimation methods have been proposed (see Materials and methods). When the tracking procedure is used, the 'mean estimation' method is still the most popular (presumably because it is very easy to use). However, the mean method is strongly influenced by the right tail (skew) of the go RT distribution (see Appendix 2 for examples), as well as by go omissions (i.e. go trials on which no response is executed). The simulations reported in Appendix 2 and summarized in *Figure 2* indicate that the integration method (which replaces go omissions with the maximum RT in order to compensate for the lacking response) is generally less biased and more reliable than the mean method when combined with the tracking procedure. Unlike the mean method, the integration method also does not assume that p(respond|signal) is exactly 0.50 (an assumption that is often not met in empirical data). Therefore, we recommend the use of the integration method (with replacement of go omissions) when non-parametric estimation methods are used. We provide software and the source code for this estimation method (and all other recommended measures; Recommendation 12).

Please note that some parametric SSRT estimation methods are less biased than even the best non-parametric methods and avoid other problems that can beset them (see *Box 2*); however, they can be harder for less technically adept researchers to use, and they may require more trials (see *Matzke et al., 2018*, for a discussion).

## Recommendation 9: Refrain from estimating SSRT when the probability of responding on stop trials deviates substantially from 0.50 or when the probability of omissions on go trials is high

Even though the preferred integration method (with replacement of go omissions) is less influenced by deviations in p(respond|signal) and go omissions than other methods, it is not completely immune to them either (*Figure 2* and Appendix 2). Previous work suggests that SSRT estimates are most reliable (*Band et al., 2003*) when probability of responding on a stop trial is relatively close to 0.50. Therefore, we recommend that researchers refrain from estimating individual SSRTs when p (respond|signal) is lower than 0.25 or higher than 0.75 (*Congdon et al., 2012*). Reliability of the estimates is also influenced by go performance. As the probability of a go omission increases, SSRT estimates also become less reliable. *Figure 2* and the resources described in Appendix 3 can be used to determine an acceptable level of go omissions at a study level. Importantly, researchers should decide on these cut-offs or exclusion criteria before data collection has started.

## How to report stop-signal experiments

## Recommendation 10: Report the methods in enough detail

To allow proper evaluation and replication of the study findings, and to facilitate follow-up studies, researchers should carefully describe the stimuli, materials, and procedures used in the study, and provide a detailed overview of the performed analyses (including a precise description of how SSRT was estimated). This information can be presented in Supplementary Materials in case of journal restrictions. *Box 3* provides a check-list that can be used by authors and reviewers. We also encourage researchers to share their software and materials (e.g. the actual stimuli).

## Recommendation 11: Report possible exclusions in enough detail

As outlined above, researchers should refrain from estimating SSRT when the independence assumptions are seriously violated or when sub-optimal task performance might otherwise compromise the reliability of the estimates. The number of participants for whom SSRT was not estimated should be clearly mentioned. Ideally, dependent variables which are directly observed (see Recommendation 12) are separately reported for the participants that are not included in the SSRT analyses. Researchers should also clearly mention any other exclusion criteria (e.g. outliers based on distributional analyses, acceptable levels of go omissions, etc.), and whether those were set a-priori (analytic plans can be preregistered on a public repository, such as the Open Science Framework; *Nosek et al., 2018*).

### Recommendation 12: Report all relevant behavioral data

Researchers should report all relevant descriptive statistics that are required to evaluate the findings of their stop-signal study (see *Box 3* for a check-list). These should be reported for each group or condition separately. As noted above (Recommendation 7), additional checks of the independent race model can be reported when the number of stop trials is sufficiently high. Finally, we encourage researchers to share their anonymized raw (single-trial) data when possible (in accordance with the FAIR data guidelines; *Wilkinson et al., 2016*).

## Conclusion

Response inhibition and impulse control are central topics in various fields of research, including neuroscience, psychiatry, psychology, neurology, pharmacology, and behavioral sciences, and the stop-signal task has become an essential tool in their study. If properly used, the task can reveal unique information about the underlying neuro-cognitive control mechanisms. By providing clear recommendations, and open-source resources, this paper aims to further increase the quality of research in the response-inhibition and impulse-control domain and to significantly accelerate its progress across the various important domains in which it is routinely applied.

## Materials and methods

The independent race model (*Box 1*) provides two common 'non-parametric' methods for estimating SSRT: the integration method and the mean method. Both methods have been used in slightly different flavors in combination with the SSD tracking procedure (see Recommendation 4). Here, we discuss the two most typical estimation variants, which we further scrutinized in our simulations (Appendix 2). We refer the reader to Appendices 2 and 3 for a detailed description of the simulations.

### Integration method (with replacement of go omissions)

In the integration method, the point at which the stop process finishes (*Box 1*) is estimated by 'integrating' the RT distribution and finding the point at which the integral equals p(respond|signal). The finishing time of the stop process corresponds to the nth RT, with n = the number of RTs in the RT distribution of go trials multiplied by p(respond|signal). When combined with the tracking procedure, overall p(respond|signal) is used. For example, when there are 200 go trials, and overall p(respond|signal) is 0.45, then the nth RT is the 90th fastest go RT. SSRT can then be estimated by subtracting mean SSD from the nth RT. To determine the nth RT, all go trials with a response are included (*including go trials with a choice error and go trials with a premature response*). Importantly, go omissions (i.e. go trials on which the participant did not respond before the response deadline) are assigned the maximum RT in order to compensate for the lacking response. Premature responses on unsuccessful stop trials (i.e. responses executed before the stop signal is presented) should also be included when calculating p(respond|signal) and mean SSD (as noted in Recommendation 4, SSD should also be adjusted after such trials). This version of the integration method produces the most reliable and least biased non-parametric SSRT estimates (Appendix 2).

### The mean method

The mean method uses the mean of the inhibition function (which describes the relationship between p(respond|signal) and SSD). Ideally, this mean corresponds to the average SSD obtained with the tracking procedure when p(respond|signal) = 0.50 (and often this is taken as a given despite some variation). In other words, the mean method assumes that the mean RT equals SSRT + mean SSD, so SSRT can be estimated easily by subtracting mean SSD from mean RT on go trials when the tracking procedure is used. The ease of use has made this the most popular estimation method. However, our simulations show that this simple version of the mean method is biased and generally less reliable than the integration method with replacement of go omissions.

## Acknowledgements

This work was mainly supported by an ERC Consolidator grant awarded to FV (European Union's Horizon 2020 research and innovation programme, grant agreement No 769595).

## Additional information

### Competing interests

Nicole C Swann: Reviewing editor, *eLife*. Adam R Aron: Reviewing editor, *eLife*. Christian Beste: has received payment for consulting and speaker's honoraria from GlaxoSmithKline, Novartis, Genzyme, and Teva. He has recent research grants with Novartis and Genzyme. Samuel R Chamberlain: consults for Shire, Ieso Digital Health, Cambridge Cognition, and Promentis. Dr Chamberlain's research is funded by Wellcome Trust (110049/Z/15/Z). Trevor W Robbins: consults for Cambridge Cognition, Mundipharma and Unilever. He receives royalties from Cambridge Cognition (CANTAB) and has recent research grants with Shionogi and SmallPharma. Katya Rubia: has received speaker's honoraria and grants for other projects from Eli Lilly and Shire. Russell J Schachar: has consulted to Highland Therapeutics, Eli Lilly and Co., and Purdue Pharma. He has commercial interest in a cognitive rehabilitation software company, eHave. The other authors declare that no competing interests exist.

### Funding

| Funder | Grant reference number | Author |
| --- | --- | --- |
| H2020 European Research Council | 769595 | Frederick Verbruggen |

The funders had no role in study design, data collection and interpretation, or the decision to submit the work for publication.

### Author contributions

Frederick Verbruggen, Conceptualization, Resources, Data curation, Software, Formal analysis, Supervision, Funding acquisition, Validation, Investigation, Visualization, Methodology, Writing—original draft, Project administration, Writing—review and editing; Adam R Aron, Christian Beste, Patrick G Bissett, Adam T Brockett, Joshua W Brown, Samuel R Chamberlain, Christopher D Chambers, Hans Colonius, Lorenza S Colzato, Brian D Corneil, James P Coxon, Annie Dupuis, Dawn M Eagle, Hugh Garavan, Ian Greenhouse, René J Huster, Sara Jahfari, J Leon Kenemans, Inge Leunissen, Chiang-Shan R Li, Dora Matzke, Sharon Morein-Zamir, Aditya Murthy, Martin Paré, Russell A Poldrack, K Richard Ridderinkhof, Trevor W Robbins, Matthew Roesch, Katya Rubia, Russell J Schachar, Jeffrey D Schall, Ann-Kathrin Stock, Nicole C Swann, Katharine N Thakkar, Maurits W van der Molen, Matthijs Vink, Jan R Wessel, Robert Whelan, Bram B Zandbelt, Conceptualization, Writing—review and editing; Guido PH Band, Andrew Heathcote, Gordon D Logan, Conceptualization, Methodology, Writing—review and editing; Luc Vermeylen, Conceptualization, Resources, Software, Writing—review and editing; C Nico Boehler, Conceptualization, Resources, Software, Formal analysis, Validation, Investigation, Visualization, Methodology, Writing—original draft, Writing—review and editing

### Author ORCIDs

Frederick Verbruggen ⓘD https://orcid.org/0000-0002-7958-0719
Adam T Brockett ⓘD http://orcid.org/0000-0001-7712-5053
Hans Colonius ⓘD http://orcid.org/0000-0002-9733-6939
Brian D Corneil ⓘD http://orcid.org/0000-0002-4702-7089
James P Coxon ⓘD http://orcid.org/0000-0003-2351-8489
Ian Greenhouse ⓘD http://orcid.org/0000-0003-1467-739X
Sara Jahfari ⓘD http://orcid.org/0000-0002-1979-589X
Russell A Poldrack ⓘD http://orcid.org/0000-0001-6755-0259

Matthew Roesch (iD) https://orcid.org/0000-0003-2854-6593
Nicole C Swann (iD) https://orcid.org/0000-0003-2463-5134
Jan R Wessel (iD) http://orcid.org/0000-0002-7298-6601
C Nico Boehler (iD) http://orcid.org/0000-0001-5963-2780

**Decision letter and Author response**
Decision letter https://doi.org/10.7554/eLife.46323.026
Author response https://doi.org/10.7554/eLife.46323.027

## Additional files

**Supplementary files**
• Transparent reporting form
DOI: https://doi.org/10.7554/eLife.46323.007

**Data availability**

The code used for the simulations and all simulated data can be found on Open Science Framework (https://osf.io/rmqaw/).

The following dataset was generated:

| Author(s) | Year | Dataset title | Dataset URL | Database and Identifier |
|---|---|---|---|---|
| Verbruggen F | 2019 | Race model simulations to determine estimation bias and reliability of SSRT estimates | https://dx.doi.org/10.17605/OSF.IO/JWSF9 | Open Science Framework, 10.17605/OSF.IO/JWSF9 |

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

## Appendix 1

DOI: https://doi.org/10.7554/eLife.46323.003

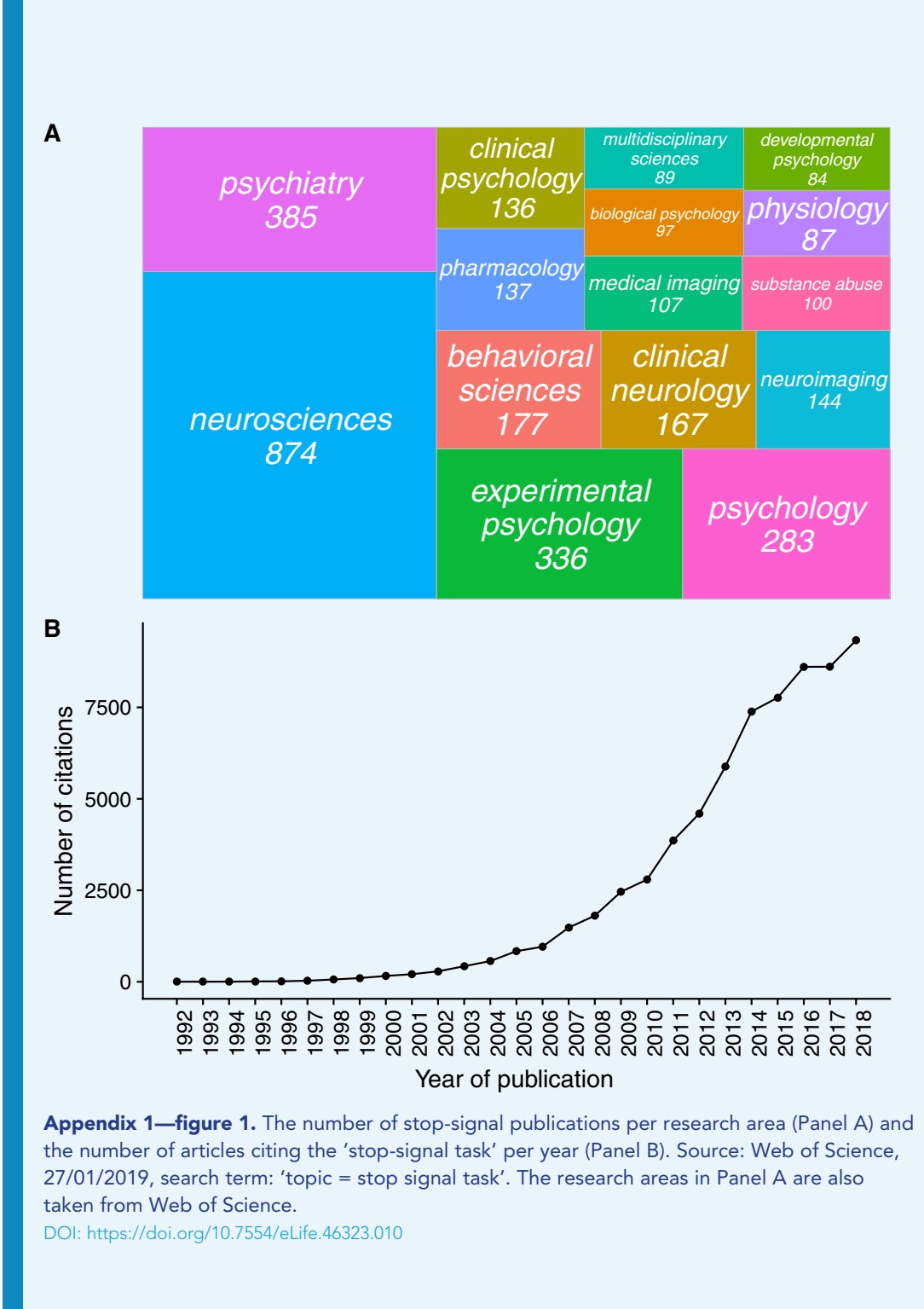

**Appendix 1—figure 1.** The number of stop-signal publications per research area (Panel A) and the number of articles citing the 'stop-signal task' per year (Panel B). Source: Web of Science, 27/01/2019, search term: 'topic = stop signal task'. The research areas in Panel A are also taken from Web of Science.

DOI: https://doi.org/10.7554/eLife.46323.010

# Appendix 2

DOI: https://doi.org/10.7554/eLife.46323.003

## Race model simulations to determine estimation bias and reliability of SSRT estimates

### Simulation procedure

To compare different SSRT estimation methods, we ran a set of simulations which simulated performance in the stop-signal task based on assumptions of the independent race model: on stop trials, a response was deemed to be stopped (successful stop) when the RT was larger than SSRT + SSD; a response was deemed to be executed (unsuccessful stop) when RT was smaller than SSRT + SSD. Go and stop were completely independent.

All simulations were done using R (**R Development Core Team, 2017**, version 3.4.2). Latencies of the go and stop runners were sampled from an ex-Gaussian distribution, using the rexGaus function (**Rigby and Stasinopoulos, 2005**, version 5.1.2). The ex-Gaussian distribution has a positively skewed unimodal shape and results from a convolution of a normal (Gaussian) distribution and an exponential distribution. It is characterized by three parameters: $\mu$ (mean of the Gaussian component), $\sigma$ (SD of Gaussian component), and $\tau$ (both the mean and SD of the exponential component). The mean of the ex-Gaussian distribution = $\mu + \tau$, and variance = $\sigma^2 + \tau^2$. Previous simulation studies of the stop-signal task also used ex-Gaussian distributions to model their reaction times (e.g. **Band et al., 2003**; **Verbruggen et al., 2013**; **Matzke et al., 2019**).

For each simulated 'participant', $\mu_{go}$ of the ex-Gaussian go RT distribution was sampled from a normal distribution with mean = 500 (i.e. the population mean) and SD = 50, with the restriction that it was larger than 300 (see **Verbruggen et al., 2013**, for a similar procedure). $\sigma_{go}$ was fixed at 50, and $\tau_{go}$ was either 1, 50, 100, 150, and 200 (resulting in increasingly skewed distributions). The RT cut-off was set at 1,500 ms. Thus, go trials with an RT >1,500 ms were considered go omissions. For some simulations, we also inserted extra go omissions, resulting in five 'go omission' conditions: 0% inserted go omissions (although the occasional go omission was still possible when $\tau_{go}$ was high), 5%, 10%, 15%, or 20%. These go omissions were randomly distributed across go and stop trials. For the 5%, 10%, 15%, and 20% go-omission conditions, we first checked if there were already go omissions due to the random sampling from the ex-Gaussian distribution. If such go omissions occurred 'naturally', fewer 'artificial' omissions were inserted.

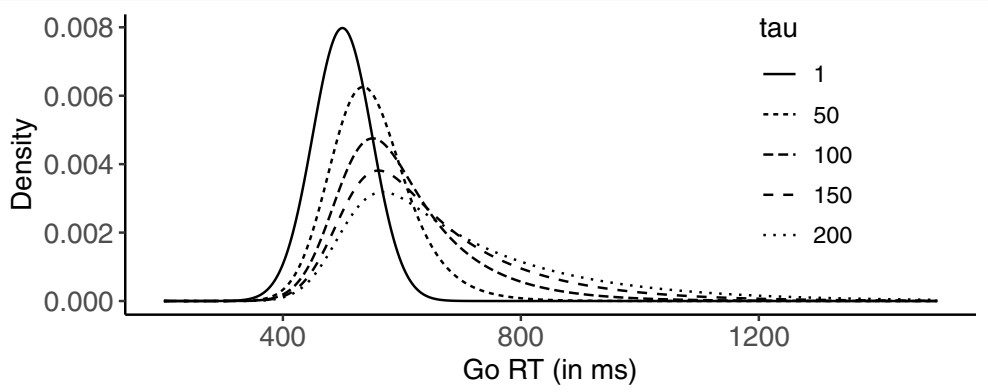

**Appendix 2—figure 1.** Examples of ex-Gaussian (RT) distributions used in our simulations. For all distributions, $\mu_{go}$ = 500 ms, and $\sigma_{go}$ = 50 ms. $\tau_{go}$ was either 1, 50, 100, 150, and 200 (resulting in increasingly skewed distributions). Note that for a given RT cut-off (1,500 ms in the simulations), cut-off-related omissions are rare, but systematically more likely as tau

increases. In addition to such 'natural' go omissions, we introduced 'artificial' ones in the different go-omission conditions of the simulations (not depicted).
DOI: https://doi.org/10.7554/eLife.46323.012

For each simulated 'participant', $\mu_{stop}$ of the ex-Gaussian SSRT distribution was sampled from a normal distribution with mean = 200 (i.e. the population mean) and SD = 20, with the restriction that it was larger than 100. $\sigma_{stop}$ and $\tau_{stop}$ were fixed at 20 and 10, respectively. For each 'participant', the start value of SSD was 300 ms, and was continuously adjusted using a standard tracking procedure (see main text) in steps of 50 ms. In the present simulations, we did not set a minimum or maximum SSD.

The total number of trials simulated per participant was either 100, 200, 400, or 800, whereas the probability of a stop signal was fixed at .25; thus, the number of stop trials was 25, 50, 100, or 200, respectively. This resulted in 5 (go omission: 0, 5, 10, 15, or 20%) x 5 ($\tau_{go}$: 1, 50, 100, 150, 200) x 4 (total number of trials: 100, 200, 400, 800) conditions. For each condition, we simulated 1000 participants. Overall, this resulted in 100,000 participants (and 375,000,000 trials).

The code used for the simulations and all simulated data can be found on Open Science Framework (https://osf.io/rmqaw/).

## Analyses

We performed three sets of analyses. First, we checked if RT on unsuccessful stop trials was numerically shorter than RT on go trials. Second, we estimated SSRTs using the two estimation methods described in the main manuscript (Materials and methods), and two other methods that have been used in the stop-signal literature. The first additional approach is a variant of the integration method described in the main manuscript. The main difference is the exclusion of go omissions (and sometimes choice errors on unsuccessful stop trials) from the go RT distribution when determining the nth RT. The second additional variant also does not assign go omissions the maximum RT. Rather, this method adjusts p(respond|signal) to compensate for go omissions (***Tannock et al., 1989***):

$$p(respond|signal)_{adjusted} = 1 - \frac{p(inhibit|signal) - p(omission|go)}{1 - p(omission|go)}$$

The nth RT is then determined using the adjusted p(respond|signal) and the distribution of RTs of all go trials with a response.

Thus, we estimated SSRT using four different methods: (1) integration method with replacement of go omissions; (2) integration method with exclusion of go omissions; (3) integration method with adjustment of p(respond|signal); and (4) the mean method. For each estimation method and condition (go omission x $\tau_{go}$ x number of trials), we calculated the difference between the estimated SSRT and the actual SSRT; positive values indicate that SSRT is overestimated, whereas negative values indicate that SSRT is underestimated. For each estimation method, we also correlated the true and estimated values across participants; higher values indicate more reliable SSRT estimates.

We investigated all four mentioned estimation approaches in the present appendix. In the main manuscript, we provide a detailed overview focussing on (1) the integration method with replacement of go omissions and (2) the mean method. As described below, the integration method with replacement of go omissions was the least biased and most reliable, but we also show the mean method in the main manuscript to further highlight the issues that arise when this (still popular) method is used.

## Results

All figures were produced using the ggplot2 package (version 3.1.0 ***Wickham, 2016***). The number of excluded 'participants' (i.e. RT on unsuccessful stop trials > RT on go trials) is presented in ***Figure 2*** of the main manuscript. Note that these are only apparent violations of

the independent race model, as go and stop were always modelled as independent runners. Instead, the longer RTs on unsuccessful stop trials result from estimation uncertainty associated with estimating mean RTs using scarce data. However, as true SSRT of all participants was known, we could nevertheless compare the SSRT bias for included and excluded participants. As can be seen in the table below, estimates were generally much more biased for 'excluded' participants than for 'included' participants. Again this indicates that extreme data are more likely to occur when the number of trials is low.

**Appendix 2—table 1.** The mean difference between estimated and true SSRT for participants who were included in the main analyses and participants who were excluded (because average RT on unsuccessful stop trials > average RT on go trials). We did this only for $\tau_{go}$ = 1 or 50, p(go omission)=10, 15, or 20, and number of trials = 100 (i.e. when the number of excluded participants was high; see Panel A, *Figure 2* of the main manuscript).

| Estimation method | Included | Excluded |
| --- | --- | --- |
| Integration with replacement of go omissions | −6.4 | −35.8 |
| Integration without replacement of go omissions | −19.4 | −48.5 |
| Integration with adjusted p(respond\|signal) | 12.5 | −17.4 |
| Mean | −16.0 | −46.34 |

DOI: https://doi.org/10.7554/eLife.46323.013

To further compare differences between estimated and true SSRTs for the included participants, we used 'violin plots'. These plots show the distribution and density of SSRT difference values. We created separate plots as a function of the total number of trials (100, 200, 400, and 800), and each plot shows the SSRT difference as a function of estimation method, percentage of go omissions, and $\tau_{go}$ (i.e. the skew of the RT distribution on go trials; see *Appendix 2—figure 1*). The plots can be found below. The first important thing to note is that the scales differ between subplots. This was done intentionally, as the distribution of difference scores was wider when the number of trials was lower (with fixed scales, it is difficult to detect meaningful differences between estimation methods and conditions for higher trial numbers; i.e. Panels C and D). In other words, low trial numbers will produce more variable and less reliable SSRT estimates.

Second, the violin plots show that SSRT estimates are strongly influenced by an increasing percentage of go omissions. The figures show that the integration method with replacement of go omissions, integration method with exclusion of go omissions, and the mean method all have a tendency to underestimate SSRT as the percentage of go omissions increases; importantly, *this underestimation bias is most pronounced for the integration method with exclusion of go omissions*. By contrast, the integration method which uses the adjusted p (respond\|signal) will overestimate SSRT when go omissions are present; compared with the other methods, this bias was the strongest in absolute terms.

Consistent with previous work (*Verbruggen et al., 2013*), skew of the RT distribution also strongly influenced the estimates. SSRT estimates were generally more variable as $\tau_{go}$ increased. When the probability of a go omission was low, the integration methods showed a small underestimation bias for high levels of $\tau_{go}$, whereas the mean method showed a clear overestimation bias for high levels of $\tau_{go}$. In absolute terms, this overestimation bias for the mean method was more pronounced than the underestimation bias for the integration methods. For higher levels of go omissions, the pattern became more complicated as the various biases started to interact. Therefore, we also correlated the true SSRT with the estimated SSRT to compare the different estimation methods.

To calculate the correlation between true and estimated SSRT for each method, we collapsed across all combinations of $\tau_{go}$, go-omission rate, and number of trials. The correlation (i.e. reliability of the estimate) was highest for the integration method with replacement of go omissions, $r$ = 0.57 (as shown in the violin plots, this was also the least biased method); intermediate for the mean method, $r$ = 0.53, and the integration method

with exclusion of go errors, $r = 0.51$; and lowest for the integration method using adjusted p (respond|signal), $r = 0.43$.

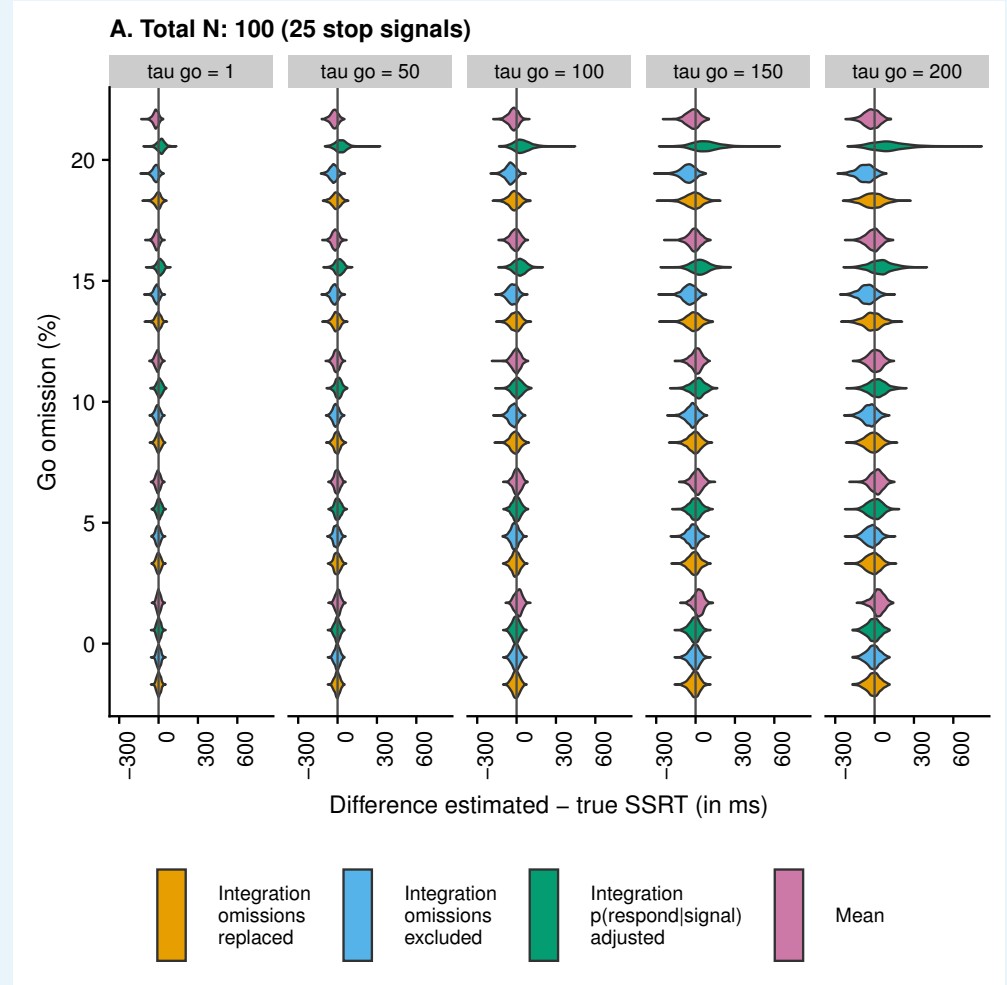

**Appendix 2—figure 2.** Violin plots showing the distribution and density of the difference scores between estimated and true SSRT as a function of condition and estimation method when the total number of trials is 100 (25 stop trials). Values smaller than zero indicate underestimation; values larger than zero indicate overestimation.

DOI: https://doi.org/10.7554/eLife.46323.014

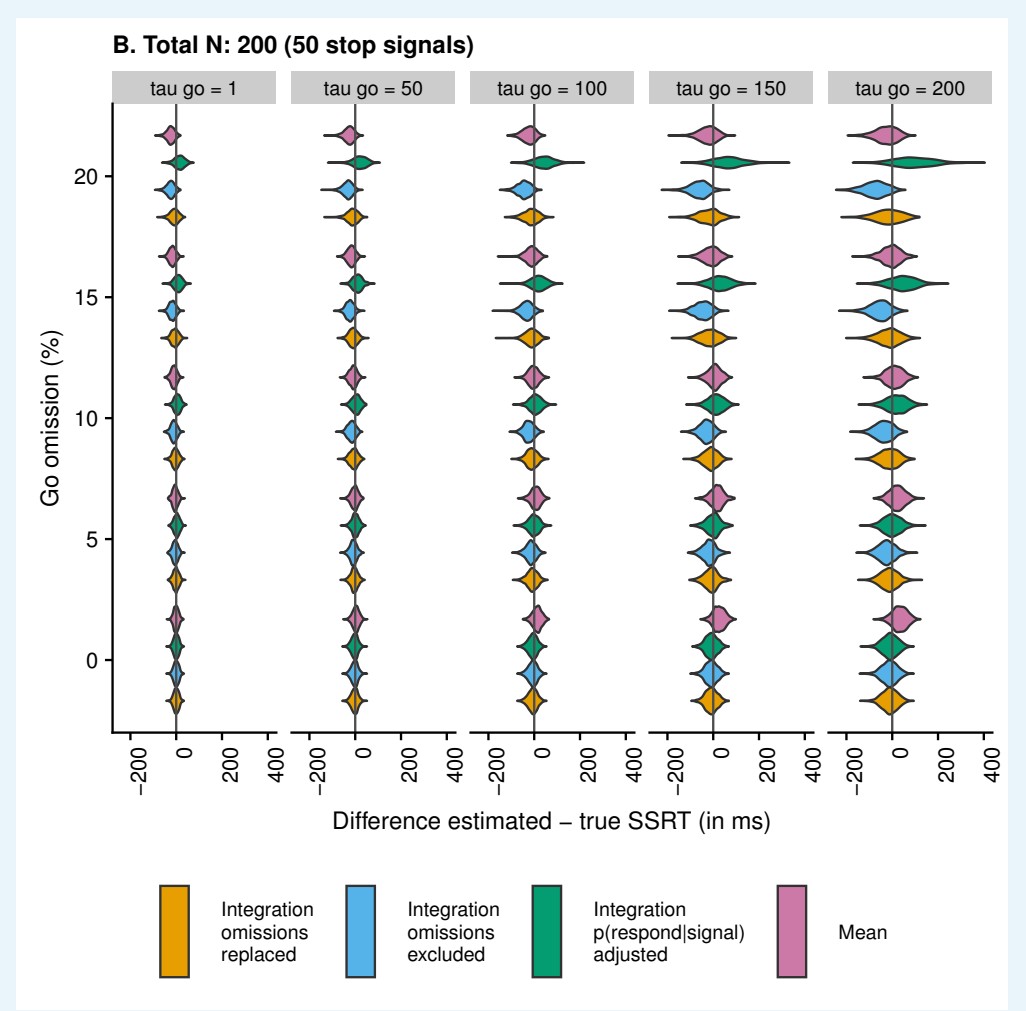

**Appendix 2—figure 3.** Violin plots showing the distribution and density of the difference scores between estimated and true SSRT as a function of condition and estimation method when the total number of trials is 200 (50 stop trials). Values smaller than zero indicate underestimation; values larger than zero indicate overestimation.

DOI: https://doi.org/10.7554/eLife.46323.015

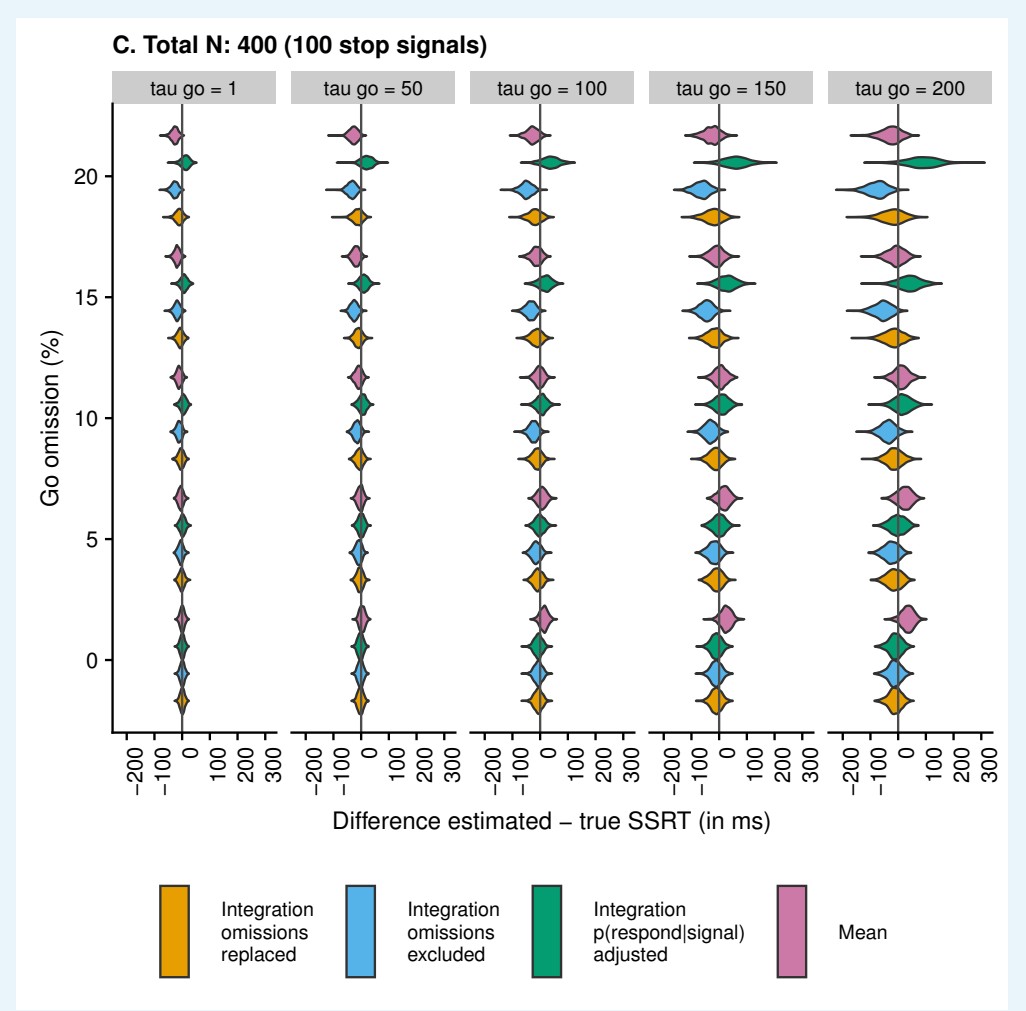

**Appendix 2—figure 4.** Violin plots showing the distribution and density of the difference scores between estimated and true SSRT as a function of condition and estimation method when the total number of trials is 400 (100 stop trials). Values smaller than zero indicate underestimation; values larger than zero indicate overestimation.

DOI: https://doi.org/10.7554/eLife.46323.016

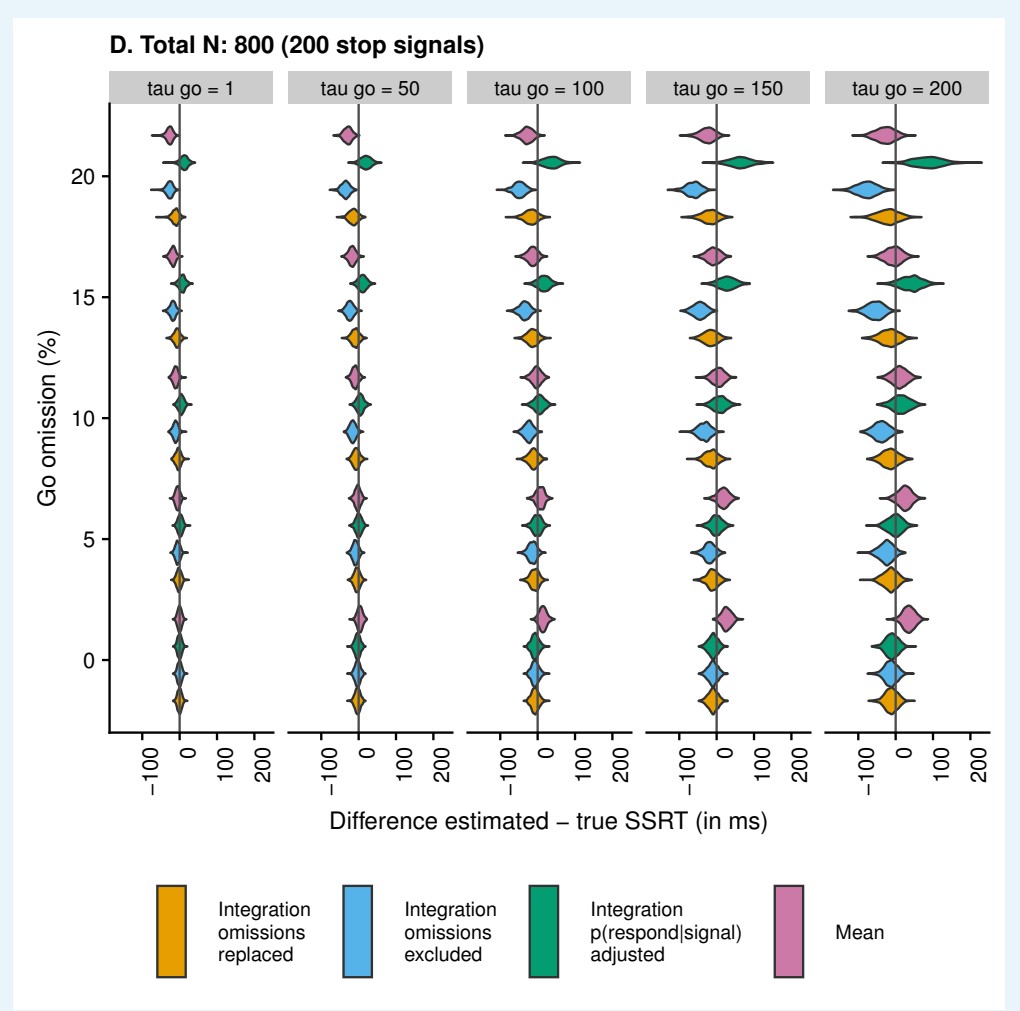

**Appendix 2—figure 5.** Violin plots showing the distribution and density of the difference scores between estimated and true SSRT as a function of condition and estimation method when the total number of trials is 800 (200 stop trials). Values smaller than zero indicate underestimation; values larger than zero indicate overestimation.
DOI: https://doi.org/10.7554/eLife.46323.017

## Appendix 3

DOI: https://doi.org/10.7554/eLife.46323.003

# Race model simulations to determine achieved power

## Simulation procedure

To determine how different parameters affected the power to detect SSRT differences, we simulated 'experiments'. We used the same general procedure as described in Appendix 2. In the example described below, we used a simple between-groups design with a control group and an experimental group.

For each simulated 'participant' of the 'control group', $\mu_{go}$ of the ex-Gaussian go RT distribution was sampled from a normal distribution with mean = 500 (i.e. the population mean) and SD = 100, with the restriction that it was larger than 300. $\sigma_{go}$ and $\tau_{go}$ were both fixed at 50, and the percentage of (artificially inserted) go omissions was 0% (see Appendix 2). $\mu_{stop}$ of the ex-Gaussian SSRT distribution was also sampled from a normal distribution with mean = 200 (i.e. the population mean) and SD = 40, with the restriction that it was larger than 100. $\sigma_{stop}$ and $\tau_{stop}$ were fixed at 20 and 10, respectively. Please note that the SDs for the population means were higher than the values used for the simulations reported in Appendix 2 to allow for extra between-subjects variation in our groups.

For the 'experimental group', the go and stop parameters could vary across 'experiments'. $\mu_{go}$ was sampled from a normal distribution with population mean = 500, 525, or 575 (SD = 100). $\sigma_{go}$ was 50, 52.5, or 57.5 (for population mean of $\mu_{go}$ = 500, 525, and 575, respectively), and $\tau_{go}$ was either 50, 75, or 125 (also for population mean of $\mu_{go}$ = 500, 525, and 575, respectively). Remember that the mean of the ex-Gaussian distribution = $\mu + \tau$ (Appendix 2). Thus, mean go RT of the experimental group was either 550 ms (500 + 50, which is the same as the control group), 600 (525 + 75), or 700 (575 + 125). The percentage of go omissions for the experimental group was either 0% (the same as the experimental group), 5% (for $\mu_{go}$ = 525) or 10% (for $\mu_{go}$ = 575).

**Appendix 3—table 1.** Parameters of the go distribution for the control group and the three experimental conditions. SSRT of all experimental groups differed from SSRT in the control group (see below).

| Parameters of go distribution | Control | Experimental 1 | Experimental 2 | Experimental 3 |
|---|---|---|---|---|
| $\mu_{go}$ | 500 | 500 | 525 | 575 |
| $\sigma_{go}$ | 50 | 50 | 52.5 | 57.5 |
| $\tau_{go}$ | 50 | 50 | 75 | 125 |
| **go omission** | 0 | 0 | 5 | 10 |

DOI: https://doi.org/10.7554/eLife.46323.019

$\mu_{stop}$ of the 'experimental-group' SSRT distribution was sampled from a normal distribution with mean = 210 or 215 (SD = 40). $\sigma_{stop}$ was 21 or 21.5 (for $\mu_{stop}$ = 210 and 215, respectively), and $\tau_{stop}$ was either 15 or 20 (for $\mu_{stop}$ = 210 and 215, respectively). Thus, mean SSRT of the experimental group was either 225 ms (210 + 15, corresponding to a medium effect size; Cohen's d $\approx$ 0.50–0.55. Note that the exact value could differ slightly between simulations as random samples were taken) or 235 (215 + 20, corresponding to a large effect size; Cohen's d $\approx$ 0.85–0.90). SSRT varied independently from the go parameters (i.e. $\mu_{go}$ + $\tau_{go}$, and % go omissions).

The total number of trials per experiment was either 100 (25 stop trials), 200 (50 stop trials) or 400 (100 stop trials). Other simulation parameters were the same as those described in Appendix 2. Overall, this resulted in 18 different combinations: 3 (go difference between control and experimental; see *Appendix 3—table 1* above) x 2 (mean SSRT difference

between control and experimental: 15 or 30) x 3 (total number of trials: 100, 200 or 400). For each parameter combination, we simulated 5000 'pairs' of subjects.

The code and results of the simulations are available via the Open Science Framework (https://osf.io/rmqaw/); stop-signal users can adjust the scripts (e.g. by changing parameters or even the design) to determine the required sample size given some consideration about the expected results. Importantly, the present simulation code provides access to a wide set of parameters (i.e. go omission, parameters of the go distribution, and parameters of the SSRT distribution) that could differ across groups or conditions.

## Analyses

SSRTs were estimated using the integration method with replacement of go omissions (i.e. the method that came out on top in the other set of simulations). Once the SSRTs were estimated, we randomly sampled 'pairs' to create the two groups for each 'experiment'. For the 'medium' SSRT difference (i.e. 210 vs. 225 ms), group size was either 32, 64, 96, 128, 160, or 192 (the total number of participants per experiment was twice the group size). For the 'large' SSRT difference (i.e. 210 vs. 235 ms), group size was either 16, 32, 48, 64, 80, or 96 (the total number of participants per experiment was twice the group size). For each sample size and parameter combination (see above), we repeated this procedure 1000 times (or 1000 experiments).

For each experiment, we subsequently compared the estimated SSRTs of the control and experiment groups with an independent-samples t-test (assuming unequal variances). Then we determined for each sample size x parameter combination the proportion of t-tests that were significant (with $\alpha$ = 0.05).

## Results

The figure below plots achieved power as a function of sample size (per group), experimental vs. control group difference in true SSRT, and group differences in go performance. Note that if true and estimated SSRTs would exactly match (i.e. estimations reliability = 1), approximately 58 participants per group would be required to detect a medium-sized true SSRT difference with power = 0.80 (i.e. when Cohen's d $\approx$ 0.525), and 22 participants per group for a large-sized true SSRT difference (Cohen's d $\approx$ 0.875).

Inspection of the figure clearly reveals that achieved power generally increases when sample size and number of trials increase. Obviously achieved power is also strongly dependent on effect size (Panel A vs. B). Interestingly, the figure also shows that the ability to detect SSRT differences is reduced when go performance of the groups differ substantially (see second and third columns of Panel A). As noted in the main manuscript and Appendix 2, even the integration method (with replacement of go omissions) is not immune to changes in the go performance. More specifically, SSRT will be underestimated when the RT distribution is skewed (note that all other approaches produce even stronger biases). In this example, the underestimation bias will reduce the observed SSRT difference (as the underestimation bias is stronger for the experimental group than for the control group). Again, this highlights the need to encourage consistent fast responding (reducing the right-end tail of the distribution).

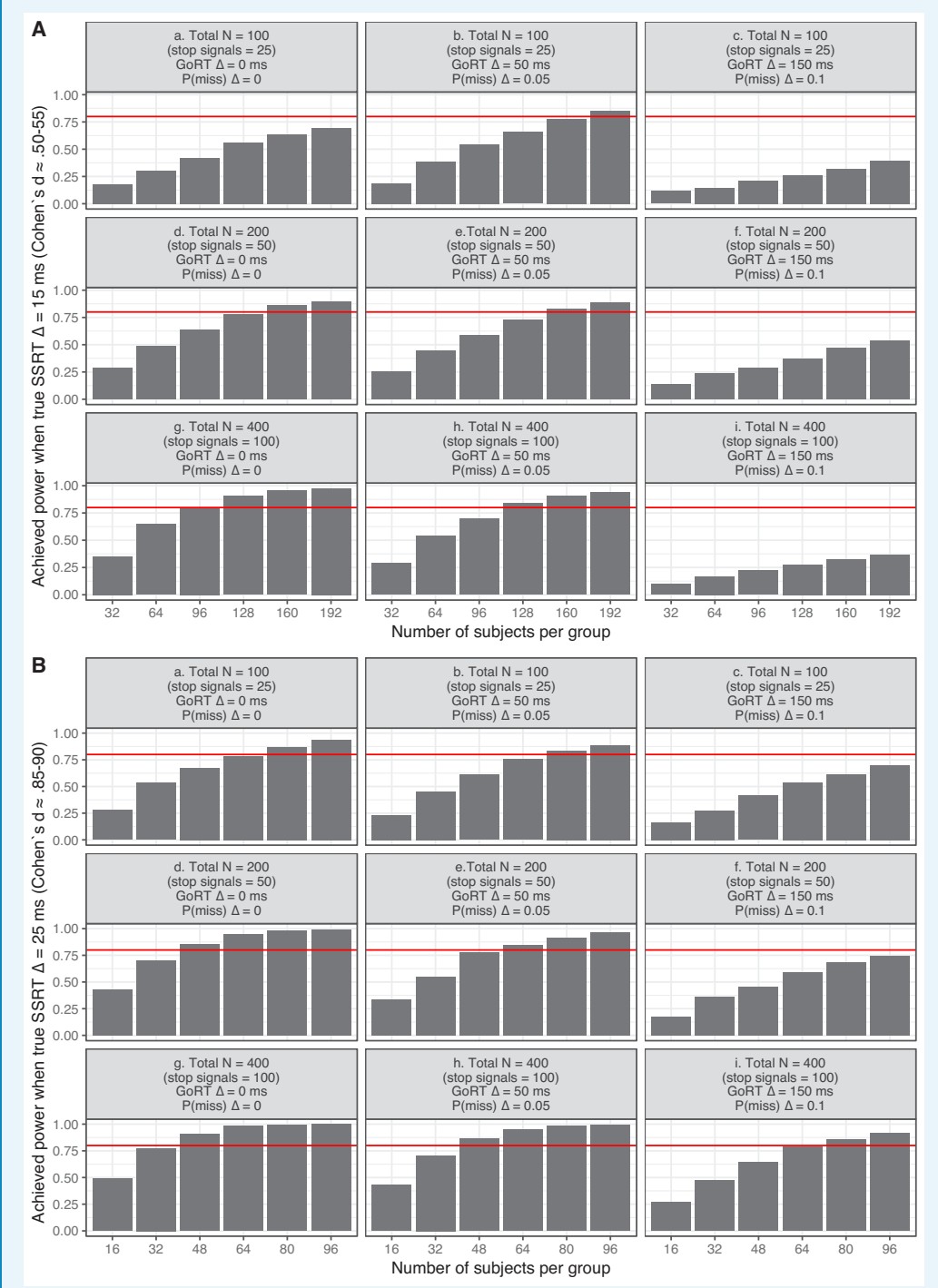

**Appendix 3—figure 1.** Achieved power for an independent two-groups design as function of differences in go omission, go distribution, SSRT distribution, and the number of trials in the 'experiments'.

DOI: https://doi.org/10.7554/eLife.46323.020

**Appendix 4**

DOI: https://doi.org/10.7554/eLife.46323.003

## Overview of the main labels and common alternatives

| Label | Description | Common alternative labels |
|---|---|---|
| Stop-signal task | A task used to measure response inhibition in the lab. Consists of a go component (e.g. a two-choice discrimination task) and a stop component (suppressing the response when an extra signal appears). | Stop-signal reaction time task, stop-signal paradigm, countermanding task |
| Go trial | On these trials (usually the majority), participants respond to the go stimulus as quickly and accurately as possible (e.g. left arrow = left key, right arrow = right key). | No-signal trial, no-stop-signal trial |
| Stop trial | On these trials (usually the minority), an extra signal is presented after a variable delay, instructing participants to stop their response to the go stimulus. | Stop-signal trial, signal trial |
| Successful stop trial | On these stop trials, the participants successfully stopped (inhibited) their go response. | Stop-success trial, signal-inhibit trial, canceled trial |
| Unsuccessful stop trial | On these stop trials, the participants could not inhibit their go response; hence, they responded despite the (stop-signal) instruction not to do so. | Stop-failure trial, signal-respond trial, noncanceled trial, stop error |
| Go omission | Go trials without a go response. | Go-omission error, misses, missed responses |
| Choice errors on go trials | Incorrect response on a go trial (e.g. the go stimulus required a left response but a right response was executed). | (Go) errors, incorrect (go or no-signal) trials |
| Premature response on a go trial | A response executed before the presentation of the go stimulus on a go trial. This can happen when go-stimulus presentation is highly predictable in time (and stimulus identity is not relevant to the go task; e.g. in a simple detection task) or when participants are 'impulsive'. Note that response latencies will be negative on such trials. | |
| p(respond\|signal) | Probability of responding on a stop trial. Non-parametric estimation methods (Materials and Methods) use p(respond\|signal) to determine SSRT. | P(respond), response rate, p(inhibit)=1 p(respond\|signal) |
| Choice errors on unsuccessful stop trials | Unsuccessful stop trials on which the incorrect go response was executed (e.g. the go stimulus required a left response but a right response was executed). | Incorrect signal-respond trials |
| Premature responses on unsuccessful stop trials | This is a special case of unsuccessful stop trials, referring to go responses executed *after* the presentation of the go stimulus but *before* the presentation of the stop signal. In some studies, this label is also used for go responses executed before the presentation of the go stimulus on stop trials (see description premature responses on go trials). | Premature signal-respond |

*continued*

| Label | Description | Common alternative labels |
|---|---|---|
| Trigger failures on stop trials | Failures to launch the stop process or 'runner' on stop trials (see **Box 2** for further discussion). | |
| Reaction time (RT) on go trials | How long does it take to respond to the stimulus on go trials? This corresponds to the finishing time of the go runner in the independent race model. | Go RT, go latency, no-signal RT |
| Stop-signal delay (SSD) | The delay between the presentation of the go stimulus and the stop signal | Stimulus-onset asynchrony (SOA) |
| Stop-signal reaction time (SSRT) | How long does it take to stop a response? SSD + SSRT correspond to the finishing time of the stop runner in the independent race model. | Stop latency |
| RT on unsuccessful stop trials | Reaction time of the go response on unsuccessful stop trials | Signal-respond RT, SR-RT (note that this abbreviation is highly similar to the abbreviation for stop-signal reaction time, which can cause confusion) |

Note: The different types of unsuccessful stop trials (including choice errors and premature responses) are usually collapsed when calculating p(respond|signal), estimating SSRT, or tracking SSD.

