## [Decision Letter]

Your article, "Capturing the ability to inhibit actions and impulsive behaviors: A consensus guide to the stop-signal task", has been reviewed by two reviewers and accepted for publication in *eLife*. This decision was overseen by Michael Frank (Senior Editor) and David Badre (Reviewing Editor).

This article provides an expert consensus view and practical guide of best practices in measuring response inhibition with the stop-signal task. It is a well-written and timely piece. The authors have done a great service for the field. The guide and resources provided will be tools of high value to both basic scientists and those in applied domains, such as clinicians interested in leveraging inhibition. The reviewers and the editors were in consensus on these strengths. There was also agreement that no major weaknesses were found in review. As *eLife* only asks revision for essential revisions, the unanimous decision was to accept the paper.

In our opinion, the paper could be made even stronger by addressing the following comments in the final submission of the paper. However, acceptance is not contingent on addressing them and they will not be reviewed again. We list these below. Congratulations on a valuable contribution to the field.

Reviewer suggestions for your consideration:

- Please provide references for this statement in the Introduction: "Research using the task has revealed links between inhibitory-control capacities and a wide range of behavioral and impulse-control problems in everyday life (e.g., attention deﬁcit/hyperactivity disorder, substance abuse, obesity, obsessive-compulsive behaviors, excessive risk-taking)." To be most accurate, this statement would need to cite studies that show a correlation between SSRT and symptom severity in the clinical populations mentioned above. This is important because SSRT may not always correlate with impulsivity (see Skippen et al., 2019).

- Trials that are called "trigger failures" (Box 2) might represent different underlying processes, depending on when the stop signal is presented in relation to the go signal. Trials in which the go and stop signals are presented simultaneously are similar to no-go trials in the go/no-go task, with trigger failures analogous to false alarm errors on no-go trials. This type of trial is thought to measure restraint, rather than cancellation (Schachar et al., 2007, 2011). Restraint (refraining from responding) and cancellation (stopping an already-prepared response, measured by SSRT) might have some distinctive neural correlates (e.g., Eagle et al., 2008; Swick et al., 2011; Sebastian et al., 2013; Zhang et al., 2017), although there is much overlap as well. Certain clinical populations could show differential impairment on restraint vs. cancellation trials. This conception of trigger failures doesn't affect any of the recommended guidelines for non-parametric estimation methods, but might be considered by the authors.

- Though the manuscript is clear and uncluttered, there was concern that the referencing may be over sparse. The authors might consider a table of key papers in this domain to aid practitioners.

- Recommendations 10-12 might be summarized in a box or table rather than bulleted in the main test.

---

## [Author Response]

Reviewer suggestions for your consideration.- Please provide references for this statement in the Introduction: "Research using the task has revealed links between inhibitory-control capacities and a wide range of behavioral and impulse-control problems in everyday life (e.g., attention deﬁcit/hyperactivity disorder, substance abuse, obesity, obsessive-compulsive behaviors, excessive risk-taking)." To be most accurate, this statement would need to cite studies that show a correlation between SSRT and symptom severity in the clinical populations mentioned above. This is important because SSRT may not always correlate with impulsivity (see Skippen et al., 2019).

Some (but not all) studies have found a correlation between SSRT and symptom severity in various clinical populations. Furthermore, SSRT correlates with the treatment outcome in some of these conditions (as noted in the manuscript). However, a full review of the clinical literature is beyond the scope of the present manuscript. Instead, we have cited a couple of systematic reviews and meta-analyses that focused on stop-signal performance in different psychopathological disorders. More specifically, in the Introduction of the revised manuscript, we now write: “Research using the task has revealed links between inhibitory-control capacities and a wide range of behavioral and impulse-control problems in everyday life, including attentiondeficit/hyperactivity disorder, substance abuse, eating disorders, and obsessive-compulsive behaviors (for meta-analyses, see e.g. Bartholdy et al., 2016; Lipszyc and Schachar, 2010; Smith et al., 2014).”

- Trials that are called "trigger failures" (Box 2) might represent different underlying processes, depending on when the stop signal is presented in relation to the go signal. Trials in which the go and stop signals are presented simultaneously are similar to no-go trials in the go/no-go task, with trigger failures analogous to false alarm errors on no-go trials. This type of trial is thought to measure restraint, rather than cancellation (Schachar et al., 2007, 2011). Restraint (refraining from responding) and cancellation (stopping an already-prepared response, measured by SSRT) might have some distinctive neural correlates (e.g., Eagle et al., 2008; Swick et al., 2011; Sebastian et al., 2013; Zhang et al., 2017), although there is much overlap as well. Certain clinical populations could show differential impairment on restraint vs. cancellation trials. This conception of trigger failures doesn't affect any of the recommended guidelines for non-parametric estimation methods, but might be considered by the authors.

We have briefly mentioned in Box 2 of the revised manuscript that different (neural) mechanisms might be involved in stopping when SSD = 0. Whether or not trigger failures at different SSDs also represent different underlying processes is an open empirical question. Note that trigger failures can occur for all SSDs (including long SSDs); therefore, we believe it is not appropriate to equate’ trigger failures’ at very short SSDs with (failures of)’ action restraint’ (at least based on our current knowledge of trigger failures).

- Though the manuscript is clear and uncluttered, there was concern that the referencing may be over sparse. The authors might consider a table of key papers in this domain to aid practitioners.

Given the main topic of the manuscript (i.e. how to empirically capture the ability to inhibit actions and impulsive behaviors in the stop-signal task), we decided to focus on key methodological papers. However, to assist readers who are not very familiar with the stop-signal literature yet, we have added in the revised version a few references to review papers that provide a detailed overview of the clinical, neuroscience, and cognitive stop-signal domains. In the Introduction of the revised manuscript, we write: “A full overview of the stop-signal literature is beyond the scope of this study (but see e.g. Aron, 2011; Bari and Robbins, 2013; Chambers et al., 2009; Schall et al., 2017; Verbruggen and Logan, 2017, for comprehensive overviews of the clinical, neuroscience, and cognitive stop-signal domains; see also the meta-analytic reviews mentioned above).”

- Recommendations 10-12 might be summarized in a box or table rather than bulleted in the main test.

As suggested by the reviewers, we have put the checklist that accompanied Recommendations 10 and 12 in a new text box, and the main text provides only a short description of the recommendations (making it more similar to the other ones). We decided to keep a summary of Recommendations 10-12 in the main text as we believe that these recommendations are equally important as all the other recommendations.